# The Quality of Statistical Reporting and Data Presentation in Predatory Dental Journals Was Lower Than in Non-Predatory Journals

**DOI:** 10.3390/e23040468

**Published:** 2021-04-16

**Authors:** Pentti Nieminen, Sergio E. Uribe

**Affiliations:** 1Medical Informatics and Data Analysis Research Group, University of Oulu, 90014 Oulu, Finland; 2Department of Conservative Dentistry and Oral Health, Riga Stradins University, LV-1007 Riga, Latvia; Sergio.uribe@rsu.lv; 3School of Dentistry, Universidad Austral de Chile, Rudloff, Valdivia 1640, Chile; 4Baltic Biomaterials Centre of Excellence, Headquarters at Riga Technical University, LV-1658 Riga, Latvia

**Keywords:** meta-research, dental research, publications, statistical reporting, data presentation

## Abstract

Proper peer review and quality of published articles are often regarded as signs of reliable scientific journals. The aim of this study was to compare whether the quality of statistical reporting and data presentation differs among articles published in ‘predatory dental journals’ and in other dental journals. We evaluated 50 articles published in ‘predatory open access (OA) journals’ and 100 clinical trials published in legitimate dental journals between 2019 and 2020. The quality of statistical reporting and data presentation of each paper was assessed on a scale from 0 (poor) to 10 (high). The mean (SD) quality score of the statistical reporting and data presentation was 2.5 (1.4) for the predatory OA journals, 4.8 (1.8) for the legitimate OA journals, and 5.6 (1.8) for the more visible dental journals. The mean values differed significantly (*p* < 0.001). The quality of statistical reporting of clinical studies published in predatory journals was found to be lower than in open access and highly cited journals. This difference in quality is a wake-up call to consume study results critically. Poor statistical reporting indicates wider general lower quality in publications where the authors and journals are less likely to be critiqued by peer review.

## 1. Introduction

Statistical reporting and presentation of data are essential parts of medical and dental research articles [1,2]. Previous studies have shown that data analysis techniques and editorial styles vary between journals, even within medical subfields [3,4]. Quantitative medical and dental research ranges from formulating study questions, planning studies, collecting material, and analysing data to reporting, interpreting, and disseminating findings. Medical articles that use statistical methods have often included poor reporting, methodological errors, and selective findings [3,5,6,7,8]. These problems in published articles are often seen as evidence that poor statistical reporting passes the peer review process. Adequate evaluation of statistical reporting and data presentation during the submission stage improves the quality of biomedical articles and journals [9,10,11,12]. Previous studies have shown that statistical information is reported more detailly, comprehensively, and usefully in eminent medical journals [13,14,15]. This is consistent with their detailed guidelines for authors, as well as a more strict review process, including extensive statistical reviewing [16]. The peer review process is undoubtedly less thorough in less visible journals [5].

Recently, Nieminen [17] introduced an instrument to evaluate the quality of statistical reporting and data presentation (SRDP) in medical articles. This tool is not specific to clinical trials or observational studies. The tool includes only items that are relevant to all studies that apply statistical methods. First, it measures the information provided in the statistical analysis (data analysis) subsection of the Material and Methods section. All quantitative papers should include full description of all data analysis methods used, verification of assumptions and software used in the analysis. Second, the tool includes items phrased as questions to evaluate the data presentation in tables and figures. Tables and figures with informative titles, clear labelling, and well-presented data are important to readers. These requirements apply to all quantitative study design (observational, experimental, trials, reliability studies, or meta-analysis). Nevertheless, application of specific data analysis methods differs between study designs [4,14].

An article published by an open access (OA) journal can be read by anyone: a researcher in the field, a scientist in another field, a student, a patient, or an interested layperson [18]. Open access publishing is a set of principles and a range of practices through which research findings are communicated online, free of purchase costs or other access barriers. Open access helps researchers as readers by opening up access to journals to which their libraries do not subscribe [19]. However, there is uncertainty and confusion about what OA entails and tensions as to how OA impacts on prestige and recognition. Major criticisms of the OA publishing have included that if OA journals have tendency to publish as many articles as possible, then quality standards may fall, and that reviewers may self-censor if their identity is not blinded [20].

Predatory publishers present themselves as academic journals to authors. However, they use lax or no peer review in their editorial processes. This is related with aggressive advertising to generate revenue from article processing charges paid by submitting authors. In this way, predatory journals exploit the OA model [21]. As peer review is the basis for reliable scholarly dissemination of research, it is the main problem presented by predatory publishers. Researchers have also started to question the validity of the term ‘predatory’ [22]. They suggest that we should stop talking about predatory publishing and start distinguishing between deceptive and low-quality journals. Researchers have also found that the available lists of “predatory” publishers and journals are not necessarily up-to-date and reliable [23].

Fast reviews and high acceptance rates are related to predatory journals [21,24]. It is possible to have a quick peer review if the manuscript is not reviewed or read at all, or if the reviewer simply skims the paper or does not require revisions [25]. Competent editors and non-statistical scientific reviewers should identify data presentation issues and substandard statistical reporting [17]. If manuscripts in predatory journals receive only superficial peer reviewing, then papers are published even if they contain significant weaknesses in statistical reporting and data presentation. The main purpose of our study was to compare the quality of data analysis reporting in highly visible subscription-based dental journals, non-predatory legitimate OA journals, and predatory OA journals. We predicted that there would be significantly lower quality of statistical reporting and data presentation in predatory journals than in non-predatory journals.

## 2. Materials and Methods

### 2.1. Set of Articles

A total of 150 articles published in 2020 were analysed, covering 3 dental journal groups: predatory OA journals, legitimate OA journals, and subscription-based visible journals. On the basis of our previous bibliometric studies, we anticipated that 50 articles per journal group would be sufficient to make comparisons between the journals. We selected only articles reporting original research findings with quantitative data published. Letters, brief reports, case reports, narrative reviews, and editorials were excluded from our sample. The sub-field of dentistry and oral health was chosen because we had previously analysed dental journals and studied the application of statistical techniques and quality of reporting in those journals [15,26,27,28].

In August 2020, we selected 50 dental articles from journals and publishers which we considered as predatory. There is no generally accepted list of predatory OA journals. We used the following criteria in deeming a journal predatory: (a) the publisher had sent invitations to submit a manuscript by email to the authors of this paper; (b) the journal fulfilled the characteristics of predatory journals described by Cobey et al. [29]; (c) the journal was not indexed by general bibliographic databases (Medline, Web of Science, Scopus); (d) the journal was not indexed by DOAJ; (e) the publisher appeared on a list of “potential, possible, or probable predatory publishers” provided by librarian Jeffrey Beall. Only journals with the presence of at least one of the following keywords listed in the name of journals were investigated: “dental”, “dentistry”, “oral”. The journals were chosen randomly from those sending invitation emails until there was a total of 50 eligible articles. To include articles from several predatory OA journals, we chose no more than 10 articles from any journal, starting with the ones most recently published. We needed to investigate 11 predatory OA journals to obtain the required 50 eligible subsequent articles for evaluation. The 11 journals are listed alphabetically in Table 1. Some journals sending invitation letters had not published any eligible articles and some only 1 article during the first 8 months of 2020.

For comparison purposes, we selected a random sample of 100 research articles reporting findings from randomised clinical trials from dental journals indexed by Medline. For the first group (legitimate OA journals), we chose randomly, with some constraints, 50 clinical trials from Medline. Included articles needed to meet the following criteria: (1) related to topics “dentistry”, “dental”, or “oral health”; (2) be a randomised clinical trial; (3) be published by a DOAJ indexed journal; (4) be available without barriers; and (5) be published in 2020.

For the second comparison group (visible subscription-based journals), we selected a random sample of 50 randomised clinical trials from 7 visible journals that are the leading journals covering dental research and have consistently been among the top 10 journals of the dentistry, oral surgery, and medicine category ranked by Garfield’s impact factor. The number of articles per journal was limited to 10. The publishing journals for the comparison articles with their impact factors in 2019 are listed alphabetically in Table 1.

We evaluated all 150 articles for their statistical reporting. We applied the checklist developed by Nieminen [17] to assess the quality of statistical reporting and data presentation (SRDP). This tool was introduced to specifically test the reporting quality of an article or submitted manuscript quickly. The applied version of the instrument included 9 items measuring the description of data procedures and reporting of findings in tables and figures. However, the tool does not assess possible statistical errors or defects. The sum of the 9 items, total quality score, measures the quality of statistical reporting and data presentation in an evaluated article. The total score ranges from 0 to 10. A published article with a very low total score (ranging from 0 to 2) indicates that the authors and peer reviewers have not focused on statistical reporting. In turn, a high score (from 9 to 10) is an initial indicator of a high-quality review and editorial process [17].

In establishing the usefulness of a measure, one must examine aspects of reliability and validity. The interobserver agreement and test–retest reliability of the Nieminen’s SRDP quality score has previously been shown to be high [17]. In applying the quality score, the critical question is: Do differences in the total score reflect the degree of reporting quality in the research reports? To demonstrate the validity of the instrument, our approach was to perform the evaluation with the quality score and one of the existing checklists in a sample of articles and see whether there was a strong correlation between the two. We chose to validate the statistical reporting quality score against the Consolidated Standards of Reporting Trials (CONSORT), a guide for authors reporting a randomised controlled trial in health research [30,31]. The CONSORT Statement is endorsed by visible general medical journals, several sub-field journals, and main editorial organisations. We used the CONSORT Statement 2010 to evaluate the reporting in the 50 randomised controlled trials published in the legitimate OA journal group. The CONSORT checklist includes 37 items and measures the overall reporting quality of clinical trials. CONSORT also includes items that assess the adherence to statistical guidelines. A researcher (SU) with long experience in evaluating papers with CONSORT read and evaluated the 50 legitimate OA articles. The first author of this paper (P.N.) evaluated the same articles using the SRDP checklist. The validity study started parallel to examining the predatory journal articles.

### 2.2. Statistical Methods

A scatter plot served for the basic visual comparison between the total scores of SRDP and CONSORT ratings among clinical trials published in the legitimate OA journals. We calculated the Pearson correlation coefficient to estimate the validity of Nieminen’s SRDP score. Cross-tabulation was used to report differences in the frequency distributions of the statistical reporting and data presentation instrument items across journal groups. The chi-squared test was employed to assess the statistically significant differences in the distributions. The mean value with standard deviation and dot plots were used to describe the distribution of the total score of SRDP by journal group. An analysis of variance was applied to estimate the statistically significant differences in the mean values of the SRDP total score. Tukey’s post hoc test [32] was used for pairwise comparisons. The Shapiro–Wilk test for normality was used to detect departures from normality. The SRDP total score variable was normally distributed, and the assumption of normality did hold for the analysis of variance test. A *p*-value < 0.05 was regarded as statistically significant. IBM SPSS Statistics version 25 (IBM Corp., Armonk, NY, USA) was used for statistical analysis.

## 3. Results

Figure 1 illustrates the agreement between the statistical reporting score and compliance of CONSORT guidelines for the clinical trials published in the legitimate OA article sample. The scatterplot indicates a consistent pattern between the two quality instruments. SRDP score well predicts articles that comply with the CONSORT guidelines. The correlation coefficient between these measurements was 0.64.

Table 2 shows the distributions of the four data presentation items of the SRDP instrument by journal group. Presentation issues in tables and figures were common in the predatory journals. Tables and figures were not prepared efficiently or accurately. The overall technical presentation of data was messy and inferior. Moreover, the general guiding principles for reporting statistical results were not followed. In legitimate OA and subscription-based visible journals, authors helped their readers better and provided more information about the study participants, variables, and statistical methods used. However, also in the visible journals, there was inadequate data reporting in over 40% of the articles.

Table 2 also compares the distribution of the sufficient description of statistical methods in the three journal groups. The articles published in the predatory journals had significant shortcomings in statistical reporting. Only 42% of these articles provided a statistical analysis subsection. An incomplete description of their statistical procedures was more common in the predatory journals than in the legitimate OA and visible subscription-based journals. Failure to name the variables and main statistical techniques for each analysis performed in the study and to verify the underlying preconditions of the main analysis methods was less common in the legitimate OA and visible journals. In addition, the legitimate OA and visible journals more often identified which statistical software had been employed.

The distribution of the SRDP quality score is shown in Figure 2. The mean (SD) quality score was 2.5 (1.4) for the predatory journals, 4.8 (1.8) for the OA journals, and 5.6 (1.8) for the more visible dental journals. The difference between the mean values was statistically significant (*p* < 0.001). The reporting quality was significantly lower in the predatory OA journal articles than in the legitimate OA dental journal article set (*p*-values for Tukey’s post hoc test was <0.001) and in the visible dental journals article set (*p*-values for Tukey’s post hoc test was <0.001). Additionally, a statistically significant difference was identified between articles published in the legitimate OA journals and articles published in the prominent dental journals (Tukey’s *p*-value = 0.044).

## 4. Discussion

We compared the reporting quality of research reports published in predatory dental journals to reports published in journals with a more legitimate status. We assessed selected data presentation characteristics and how statistical techniques were reported in the published articles. Our findings show that articles published in predatory OA dental journals reported statistical methods and presented data with lower quality than articles in the selected legitimate OA journals and prominent dental journals. The higher quality of statistical reporting was best secured in the visible dental publications. The poor data presentation of quantitative findings and insufficient description of data analysis techniques in predatory OA journals point to an issue of peer review. The papers were not adequately assessed during peer review and editorial processes, or not reviewed at all. There is a need to develop interventions and education to protect readers and authors from these low-quality journals.

The SRDP checklist also demonstrated high validity. A low statistical reporting score could indicate papers with an overall poor quality of writing and low adherence to general reporting guidelines. A high score refers to a readable and understandable paper where the authors have also focused on all aspects of reporting. This high validity of the SRDP instrument encourages its use by peer reviewers, editors, and researchers with expertise in various areas. We recognise that more work is needed to assess its applicability in other sub-fields.

Recent literature reviews show that only a few studies of predatory journals have been published that have reported empirically derived characteristics or traits of predatory journals [25,26]. These scoping reviews summarised the literature on predatory journals and concluded that research demonstrated poor quality standards in predatory journals. Moher et al. [27] examined 1907 primary biomedical articles published in more than 200 supposed predatory OA journals. Articles in their sample failed to report information necessary for readers to reproduce the results and critically evaluate the findings. The only published empirical quantitative study we could find that sheds light on the statistical reporting of predatory journals is the study by McCutcheon et al. [28]. They reported that articles published in predatory social science journals contain more statistical errors than articles published in “non-predatory” journals.

Most of studies published by predatory journals may well be serious research but inadequately written, irrelevant, or both. Some published papers may even be excellent, but young, inexperienced junior researchers looking for a fast submission process have fallen prey to a pirate publisher. Predatory journals have shortened the submission and review processes to encourage authors to send them papers. Moreover, because manuscripts have not been carefully scrutinised, authors encounter few criticisms and high acceptance rates, leading to greater author satisfaction [25].

Reviewers are responsible for evaluating the quality of research submitted for publication. They should ensure that research is relevant and conducted properly, and that the statistical presentation is good enough for readers to identify what data analysis methods have been applied and what findings are reported in the tables and figures [3,17,33,34]. The reporting of statistical information was not valuable, comprehensive, or helpful for the reader in the predatory OA journals. Our results provide important evidence of the lax review process with non-existent statistical reviewing in predatory journals. About 80% of the articles had several presentation issues. Over 70% of the tables and figures were difficult to understand due to an insufficient description of the summary statistics, tests, and methods used. Hence, a considerable number of these published articles contained obvious structural and presentation errors that reveal acceptance without a proper peer-review process or after a brief evaluation period. We found that compliance with standard guidelines for displaying information in tables and figures [35] had been required in more detail in the legitimate comparison journals.

The nature of the studies varied between the predatory journals (‘dental articles’) and the others (reports from dental RCTs). For legitimate OA and visible dental journals, we limited our article search to RCTs to ensure proper random samples from Medline and visible dental journals. The selected predatory journals did not publish many RCTs during 2020. Thus, we selected all recently published quantitative articles. This difference might explain some of the reported effects. However, the evaluation tool did not include any items that are only specific to clinical trials or observational studies. High-quality statistical reporting and data presentation are not related to the study design. Criteria pertaining to the description of statistical procedures and the reporting of results in tables and figures are the same in all articles that are essentially statistical in character. To our knowledge, there is no evidence that the statistical reporting quality of RCTs does differ from observational studies.

Five questions in the SRDP form evaluate the description of data analysis methods in the Materials and Methods section. To some researchers, these items seem commonsensical. Medical reporting guidelines recommend that quantitative research articles include a statistical analysis (or data analysis) subsection with a clear subheading in the Materials and Methods section [36,37]. However, in our experience, this is not obvious to all early-career biomedical or health science researchers. As consulting biostatisticians, we have faced this problem frequently. Our study shows that this required section was included in only 42% of the predatory articles, while it was provided in almost all the articles in the comparison groups. It is impossible to write relevant and rigorous review reports if an extended methods section is not included in submitted manuscripts. Our data support the claims that there is a lack of proper peer review in predatory journals [20,21,29,38].

## 5. Conclusions

In conclusion, we demonstrated the low quality of statistical reporting and data presentation of potential predatory journals in the dental field. This raises concerns about other quality standards in journals that are not mission-critical. Poor statistical reporting indicates wider general tolerance for poor study design, writing, and research in publications where the authors and journals are less likely to be criticised by peer review. The authors should be aware of these journals so as not to be fooled by the high acceptance rates and the lack of critiques. The readers should be aware that those articles that do not undergo a peer review process or are poorly reviewed may harm current and future scientific studies in general. Taking active measures to avoid selecting these predatory journals is the main point of limiting their spread and preventing misleading science.

## Figures and Tables

**Figure 1 entropy-23-00468-f001:**
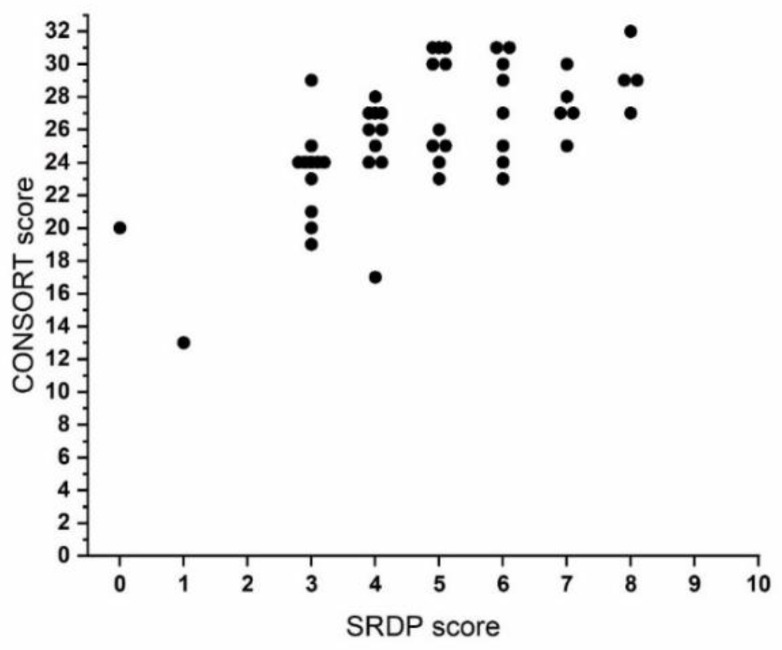
Scatter plot of statistical reporting and data presentation (SRDP) score with CONSORT score in 50 clinical trial articles published in dental open access journals.

**Figure 2 entropy-23-00468-f002:**
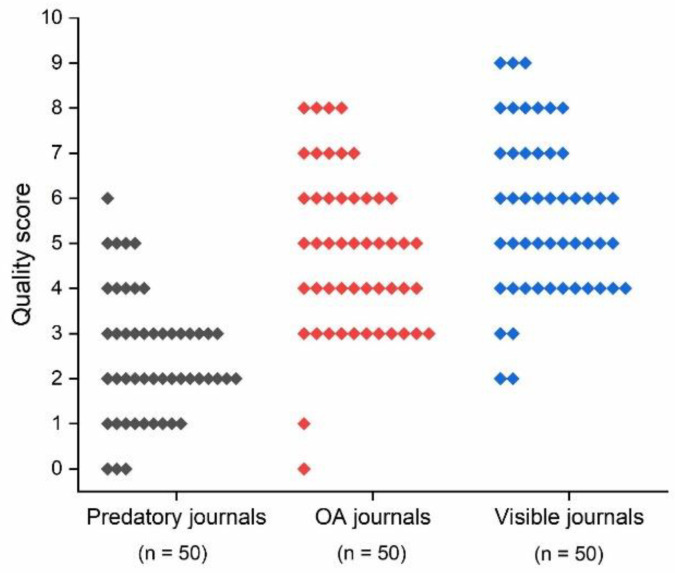
Distribution of statistical reporting and data presentation quality score (SRDP) by journal group.

**Table 1 entropy-23-00468-t001:** Dental journals surveyed for the use of statistical reporting and data presentation.

Journal	Impact Factor 2019	Number of Articles
Predatory open access (OA) journals		
*Dental Oral Biology and Craniofacial Research* (DOBCR)	-	8
*Dental, Oral and Maxillofacial Research* (DOMR)	-	1
*Dental Research and Oral Health* (DROH)	-	5
*EC Dental Science* (EC DS)	-	9
*GJMR: (J) Dentistry and Otolaryngology* (GJMR_J_DO)	-	10
*International Journal of Dentistry and Oral Health* (IJDOH)	-	10
*Journal of Dentistry and Oral Care Medicine* (JDOCM)	-	1
*Journal of Dentistry and Oral Health* (JDOH)	-	1
*JSM Dentistry* (JSMD)	-	2
*Modern Research in Dentistry* (MRD)	-	1
*Scientific Journal of Research in Dentistry* (SJRD)	-	2
Legitimate open access (OA) journals		
*Archives of Oral Biology*	1.931	1
*BDJ Open*	1.306	1
*BMC Medical Education*	1.831	1
*BMC Oral Health*	1.911	8
*Brazilian Dental Journal*	-	1
*Brazilian Oral Research*	1.633	6
*Clinical and Experimental Dental Research*	-	3
*Dental Press Journal of Orthodontics*	-	1
*Dental Materials Journal*	1.359	2
*Dentistry Journal*	-	3
*European Journal of Paediatric Dentistry*	1.500	3
*Indian Journal of Dental Research*	-	3
*International Journal of Dental Hygiene*	1.229	1
*Journal of Applied Oral Science*	1.797	4
*Journal of Indian Society of Pedodontics and Preventive Dentistry*	-	2
*Journal of Oral Science*	1.200	1
*Journal of Prosthodontic Research*	2.662	1
*Medicina Oral Patologia Oral y Cirugia Bucal*	1.596	5
*Progress in Orthodontics*	1.822	2
*Swiss Dental Journal*	-	1
Visible subscription-based dental journals		
*Clinical Implant Dentistry and Related Research* (CIDRR)	3.396	5
*Clinical Oral Implants Research* (COIR)	3.723	10
*International Endodontic Journal* (IEJ)	3.801	10
*Journal of Clinical Periodontology* (JCP)	5.241	10
*Journal of Dental Research* (JDR)	4.914	3
*Journal of Periodontology* (JP)	3.742	10
*Oral Oncology* (OO)	3.979	2

**Table 2 entropy-23-00468-t002:** The distributions of the data analysis and statistical reporting items by journal group.

	Journal Group				
Item	Predatory OA n (%)	Legitimate OA n (%)	Visible Subscription-Based n (%)	All n (%)	*p*-Value of the Chi-Squared Test
Total number of articles	50	50	50	150	
Tables and figures in the results section:					
Basic characteristics reported in a table	18 (36.0)	26 (52.0)	30 (60.0)	74 (49.3)	0.051
Total number of participants provided	11 (22.0)	7 (14.0)	15 (30.0)	33 (22.0)	0.171
Statistics, tests, and methods identified	14 (28.0)	21 (42.0)	24 (48.0)	59 (39.3)	0.110
Presentation issues:					
Several	40 (80.0)	22 (44.0)	8 (16.0)	70 (46.7)	<0.001
50% or less	9 (18.0)	12 (24.0)	17 (34.0)	38 (25.3)	
No issues	1 (2.0)	16 (32.0)	25 (50.0)	42 (28.0)	
Materials and Methods section					
Statistical analysis subsection provided	21 (42.0)	47 (94.0)	49 (98.0)	117 (78.0)	<0.001
Variables with methods identified	13 (26.0)	26 (52.0)	22 (44.0)	61 (40.7)	0.029
Assumptions verified	7 (14.0)	28 (56.0)	22 (44.0)	57 (38.0)	<0.001
References to statistical literature	0 (0.0)	0 (0.0)	10 (20.0)	10 (6.7)	<0.001
Software reported	28 (56.0)	42 (84.0)	42 (84.0)	112 (74.7)	0.001

## Data Availability

The data presented in this study are available at https://doi.org/10.5281/zenodo.4556988 (accessed on 12 March 2021).

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
