# Peer review of "The Quality of Statistical Reporting and Data Presentation in Predatory Dental Journals Was Lower Than in Non-Predatory Journals"

_entropy, 2021, doi:10.3390/e23040468_

Round 1
Reviewer 1 Report
This is an interesting and meaningful study. I only two minor concerns:
1) The names of those three journal groups should be: predatory OA journals, legitimate OA journals, subscription-based journals;
2) The comparison between observational studies (from predatory group) and RCTs (from other groups) is problematic, this should be clearly stated as a limitation of this study. It would be helpful to remind readers that CONSORT applies to RCTs but not observational studies which need to follow STROBE guidelines.
Author Response
This is an interesting and meaningful study. I only two minor concerns:
Thank you.
1) The names of those three journal groups should be: predatory OA journals, legitimate OA journals, subscription-based journals;
We have now used these terms.
2) The comparison between observational studies (from predatory group) and RCTs (from other groups) is problematic, this should be clearly stated as a limitation of this study. It would be helpful to remind readers that CONSORT applies to RCTs but not observational studies which need to follow STROBE guidelines.
We agree that that our comparison of observational vs RCTs articles may be confusing. We have now clarified that the SRDP checklist includes only items that that are not specific to study designs. It evaluates reporting characteristics that should be present in all articles that are statistical in nature. We have included the following text in the introduction:
“Recently, Nieminen introduced an instrument to evaluate the quality of statistical reporting and data presentation (SRDP) in medical articles. The tool is not specific to clinical trials or observational studies. The tool includes only items that are relevant to all papers that apply statistical methods. First, it measures the information provided in the Statistical analysis (data analysis) subsection of the Material and Methods section. All quantitative papers should include full description of all data analysis methods used, verification of assumptions and software used in the analysis. Second, the tool includes items phrased as questions to evaluate the data presentation in tables and figures. Tables and figures with informative titles, clear labelling, and well-presented data are important to readers. These requirements apply to all quantitative study design (observational, experimental, trials, reliability studies or meta-analysis).”
We have also included the following text in the discussion section to further clarify the use of the statistical reporting instrument for both observational studies and RCTs:
“This difference might explain some of the reported effects. However, the evaluation tool did not include any items that are only specific to clinical trials or observational studies. High quality statistical reporting and data presentation are not related to the study de-sign. Criteria pertaining to the description of statistical procedures and the reporting of results in tables and figures are same in all articles that are essentially statistical in character. To our knowledge, there is no evidence that the statistical reporting quality of RCTs does differ from observational studies.”
We note in the methods section that CONSORT was used only for the 50 randomized controlled trials published in the legitimate OA journal group. The purpose of this additional analysis using a sub-group of articles was to demonstrate the validity of the SRDP instrument. CONSORT was not applied for the whole article set including observational studies published in the predatory journals.
Reviewer 2 Report
It is not at all clear to me whether the sample is stratified or not (on the first layer is the diary and on the second article etc.) In the text of this paper, the authors refer sometimes to the random selection of journals and to that of articles. It is necessary to clarify and possibly explain the determination of the sample volume, in my opinion. As a statistician, if this is not done I cannot believe in the quality of the results and these become simple opinions without scientific coverage even seemingly tested. Why 100 articles or why 50, why the limit of 10 for the same publication ...? Is the sampling stratified or not? The key to this whole article is in the random sampling, written in the text and detailed by its authors, I think so ... Please try to detail it in a better way and send the final paper to be published...
Figure 1 and 2 are very large even for an old man like me who wears glasses for zooming in on the image (with + 3.5-4 for approaching images and the size of the visual validation is not essential to prove something is right, neither for myself nor for your readers, if you accept this kind of humor... bigger ideas are specific to smaller graphs)... I know that backstage of your paper there is solid research, but you must convince the audience that your sample is randomly extracted detailing the technique very precisely.
Author Response
It is not at all clear to me whether the sample is stratified or not (on the first layer is the diary and on the second article etc.) In the text of this paper, the authors refer sometimes to the random selection of journals and to that of articles. It is necessary to clarify and possibly explain the determination of the sample volume, in my opinion. As a statistician, if this is not done I cannot believe in the quality of the results and these become simple opinions without scientific coverage even seemingly tested.
We have clarified arguments justifying the selection of evaluated articles. We have also added the following text to the Methods section (2.1. Set of articles):
“The journals were chosen randomly from those sending invitation emails, until there would be a total of 50 eligible articles. To include articles from several predatory OA journals, no more than ten articles were chosen from any journal, starting with the ones most recently published. We needed to investigate 11 predatory OA journals to get the required 50 eligible subsequent articles for evaluation. The 11 journals are listed alphabetically in Table 1. Some journals sending invitation letters had not published any eligible articles and some only one article during the first eight months of 2020.”
Accordingly, we think our choice of journals and articles enables us to make externally valid statements about quality of statistical reporting and data presentation in dental journals.
Why 100 articles or why 50, why the limit of 10 for the same publication ...? Is the sampling stratified or not? The key to this whole article is in the random sampling, written in the text and detailed by its authors, I think so ... Please try to detail it in a better way and send the final paper to be published...
We have revised the methods to include a description of how we obtained the articles. We have also clarified arguments justifying our sampling (article selection). We note in the section 2.1. that “Based on our previous bibliometric studies, we anticipated that 50 articles per journal group would be sufficient to make comparisons between the journals.” In those previous studies, we have estimated using sample size calculations that a minimum of 40 articles is required per journal group.
Number of articles was limited to 10 per journal in order to include several journals in the evaluation. Thus, we would not evaluate articles mainly from one journal, or just from two or three journals or publishers. We have now clarified this point in the methods section as follows: “To include articles from several predatory OA journals, no more than ten articles were chosen from any journal, starting with the ones most recently published.”
Figure 1 and 2 are very large even for an old man like me who wears glasses for zooming in on the image (with + 3.5-4 for approaching images and the size of the visual validation is not essential to prove something is right, neither for myself nor for your readers, if you accept this kind of humor... bigger ideas are specific to smaller graphs)...
We have now reduced the size of the figures in the manuscript. We let the technical editor of the journal to decide the final size of the images.
I know that backstage of your paper there is solid research, but you must convince the audience that your sample is randomly extracted detailing the technique very precisely.
We thank the reviewer for the constructive comment.
Reviewer 3 Report
The authors investigated the quality of statistical reporting and data presentation in predatory vs non predatory scientific journals. The topic is of moderate interest for the scientific community because IF and journal ranking are already sufficient estimator with this intrinsic significance. Otherwise, it is better in any case to underline the concept “Melius abundare quam deficere” latin says. The same approach was used for my review of the manuscript.
Unfortunately, the current version of MS has several shortcomings that prevent a full appreciation of its (possible) significance.
General
The MS would benefit from minor edits to improve the readability for the readers. Some sentences are non-logical or confusing and they would need to be rewritten, e.g. The mean (SD) quality score of the statistical reporting and data presentation was 2.5 (1.4) for the predatory journals, 4.8 (1.8) for the OA journals it was and 5.6 (1.8) for the more visible dental journals.
Moreover, a strong bias is present. It is not clear why the authors have taken two different groups (RCT and non RCT) in non predatory vs predatory journals because there is a different mode to express the data.
Abstract - clear and summarizes the main idea of the manuscript.
Main text
The introduction is long and tedious. Please simplify it.Some sentences are more suitable for discussion, e.g. lines 69-78. Language about predatory journals seems a bit harsh, both in introduction and in discussion. Authors use terms that are not suitable for scientific publication, such as “they fool only minimal number”; “studies published are fake” “enjoy no reputation in science”.
Introduction could more focus on what are differences in statistical reporting then on what are predatory journals.
The methodology is correct, and it is sufficiently explained in the manuscript.
1.statistical methods: please report how normal distribution was checked
- statistical methods:please report what p value was considered statistically significant
- Table 2, please provide the N for each group in the the first row instead of last
Please underline that also in the top quality group there is a 40% of inadequate data reporting.
Final opinion:
Idea of this manuscript is good, however, the language that was used does not fit in scientific journals. AUthors use too much of introduction and discussion to present how predatory journals are bad for scientific community instead of explaining what poor statistical reporting may mean. Authors write in their discussion “Poor statistical reporting indicates wider general tolerance for poor study design, writing and research in publications where the authors and journals are less likely to be criticized by peer review.”. In my opinion, that should be the main focus of the manuscript and should be explained more in details.
Author Response
The authors investigated the quality of statistical reporting and data presentation in predatory vs non predatory scientific journals. The topic is of moderate interest for the scientific community because IF and journal ranking are already sufficient estimator with this intrinsic significance. Otherwise, it is better in any case to underline the concept “Melius abundare quam deficere” latin says. The same approach was used for my review of the manuscript.
Thank you for your kind approach to review our manuscript. We have clarified arguments justifying the topic of our study.
Unfortunately, the current version of MS has several shortcomings that prevent a full appreciation of its (possible) significance. The MS would benefit from minor edits to improve the readability for the readers. Some sentences are non-logical or confusing and they would need to be rewritten, e.g. The mean (SD) quality score of the statistical reporting and data presentation was 2.5 (1.4) for the predatory journals, 4.8 (1.8) for the OA journals it was and 5.6 (1.8) for the more visible dental journals.
Thank you. We have worked on making the flow of findings clearer.
Moreover, a strong bias is present. It is not clear why the authors have taken two different groups (RCT and non RCT) in non predatory vs predatory journals because there is a different mode to express the data.
We agree that that our comparison of observational vs RCTs articles may be confusing. We have now clarified that the SRDP checklist includes only items that that are not specific to study designs. It evaluates reporting characteristics that should be present in all articles that are statistical in nature. We have included the following text in the introduction:
“Recently, Nieminen introduced an instrument to evaluate the quality of statistical reporting and data presentation (SRDP) in medical articles. The tool is not specific to clinical trials or observational studies. The tool includes only items that are relevant to all papers that apply statistical methods. First, it measures the information provided in the Statistical analysis (data analysis) subsection of the Material and Methods section. All quantitative papers should include full description of all data analysis methods used, verification of assumptions and software used in the analysis. Second, the tool includes items phrased as questions to evaluate the data presentation in tables and figures. Tables and figures with informative titles, clear labelling, and well-presented data are important to readers. These requirements apply to all quantitative study design (observational, experimental, trials, reliability studies or meta-analysis).”
We have also included the following text in the discussion to further clarify the use of the statistical reporting instrument for both observational studies and RCTs:
“This difference might explain some of the reported effects. However, the evaluation tool did not include any items that are only specific to clinical trials or observational studies. High quality statistical reporting and data presentation are not related to the study de-sign. Criteria pertaining to the description of statistical procedures and the reporting of results in tables and figures are same in all articles that are essentially statistical in character. To our knowledge, there is no evidence that the statistical reporting quality of RCTs does differ from observational studies.”
Abstract - clear and summarizes the main idea of the manuscript.
Thank you.
The introduction is long and tedious. Please simplify it. Some sentences are more suitable for discussion, e.g. lines 69-78. Language about predatory journals seems a bit harsh, both in introduction and in discussion. Authors use terms that are not suitable for scientific publication, such as “they fool only minimal number”; “studies published are fake” “enjoy no reputation in science”.
We have now modified the manuscript accordingly. We have removed or reformulated sentences that included unsuitable terms.
Introduction could more focus on what are differences in statistical reporting then on what are predatory journals.
Thank you for drawing our attention to this issue. We have now reduced text related to the predatory publishing and added the following text:
“Recently, Nieminen introduced an instrument to evaluate the quality of statistical reporting and data presentation (SRDP) in medical articles. The tool is not specific to clinical trials or observational studies. The tool includes only items that are relevant to all papers that apply statistical methods. First, it measures the information provided in the Statistical analysis (data analysis) subsection of the Material and Methods section. All quantitative papers should include full description of all data analysis methods used, verification of assumptions and software used in the analysis. Second, the tool includes items phrased as questions to evaluate the data presentation in tables and figures. Tables and figures with informative titles, clear labelling, and well-presented data are important to readers. These requirements apply to all quantitative study design (observational, experimental, trials, reliability or meta-analysis). Nevertheless, applica-tion of specific data analysis methods differs between study designs.”
The methodology is correct, and it is sufficiently explained in the manuscript.
Thank you.
- statistical methods: please report how normal distribution was checked.
We have now reported Shapiro-Wilks test for normality in the Statistical analysis sub-section.
- statistical methods: please report what p value was considered statistically significant
We have added the following sentence to define statistical significance: “A p-value <0.05 was regarded as statistically significant.”
- Table 2, please provide the N for each group in the the first row instead of last
We have replaced the total number of articles as the first row in Table 2.
Please underline that also in the top quality group there is a 40% of inadequate data reporting.
Thank you for this comment. We now note that “However, also in the visible journals there was inadequate data reporting in over 40% of the articles.”
Idea of this manuscript is good, however, the language that was used does not fit in scientific journals. AUthors use too much of introduction and discussion to present how predatory journals are bad for scientific community instead of explaining what poor statistical reporting may mean. Authors write in their discussion “Poor statistical reporting indicates wider general tolerance for poor study design, writing and research in publications where the authors and journals are less likely to be criticized by peer review.”. In my opinion, that should be the main focus of the manuscript and should be explained more in details.
Thank you, we agree that the tone was not suitable for a scientific article in some sentences. We have edited those parts. We have also focused more on the statistical reporting and data presentation and removed text related to the concept of predatory journals.
Round 2
Reviewer 2 Report
The paper can be published now.
Reviewer 3 Report
In statistical methods was reported that boxplot was used to describe SCORE's distribution but dotplot was presented